# Rethinking the Smaller-Norm-Less-Informative Assumption in Channel Pruning of Convolution Layers

**Jianbo Ye**[*]
College of Information Sciences and Technology
The Pennsylvania State University
jxy198@ist.psu.edu

**Xin Lu, Zhe Lin**
Adobe Research
{xinl,zlin}@adobe.com

**James Z. Wang**
College of Information Sciences and Technology
The Pennsylvania State University
jwang@ist.psu.edu

## Abstract

Model pruning has become a useful technique that improves the computational efficiency of deep learning, making it possible to deploy solutions in resource-limited scenarios. A widely-used practice in relevant work assumes that a smaller-norm parameter or feature plays a less informative role at the inference time. In this paper, we propose a channel pruning technique for accelerating the computations of deep convolutional neural networks (CNNs) that does not critically rely on this assumption. Instead, it focuses on direct simplification of the channel-to-channel computation graph of a CNN without the need of performing a computationally difficult and not-always-useful task of making high-dimensional tensors of CNN structured sparse. Our approach takes two stages: first to adopt an end-to-end stochastic training method that eventually forces the outputs of some channels to be constant, and then to prune those constant channels from the original neural network by adjusting the biases of their impacting layers such that the resulting compact model can be quickly fine-tuned. Our approach is mathematically appealing from an optimization perspective and easy to reproduce. We experimented our approach through several image learning benchmarks and demonstrate its interesting aspects and competitive performance.

## 1 Introduction

Not all computations in a deep neural network are of equal importance. In a typical deep learning pipeline, an expert crafts a neural architecture, which is trained using a prepared dataset. The success of training a deep model often requires trial and error, and such loop usually has little control on prioritizing the computations happening in the neural network. Recently researchers started to develop model-simplification methods for convolutional neural networks (CNNs), bearing in mind that some computations are indeed non-critical or redundant and hence can be safely removed from a trained model without substantially degrading the model's performance. Such methods not only accelerate computational efficiency but also possibly alleviate the model's overfitting effects.

Discovering which subsets of the computations of a trained CNN are more reasonable to prune, however, is nontrivial. Existing methods can be categorized from either the *learning perspective* or from the *computational perspective*. From the learning perspective, some methods use a data-independent approach where the training data does not assist in determining which part of a trained CNN should be pruned, *e.g.* He et al. (2017) and Zhang et al. (2016), while others use a data-dependent approach through typically a joint optimization in generating pruning decisions, *e.g.*, Han et al. (2015) and Anwar et al. (2017). From the computational perspective, while most approaches

---

[*]The research was done when J. Ye was an intern at Adobe in the summer of 2017.

focus on setting the dense weights of convolutions or linear maps to be structured sparse, we propose here a method adopting a new conception to achieve in effect the same goal.

Instead of regarding the computations of a CNN as a collection of separate computations sitting at different layers, we view it as a network flow that delivers information from the input to the output through different channels across different layers. We believe saving computations of a CNN is not only about reducing what are calculated in an individual layer, but perhaps more importantly also about understanding how each channel is contributing to the entire information flow in the underlying passing graph as well as removing channels that are less responsible to such process. Inspired by this new conception, we propose to design a "gate" at each channel of a CNN, controlling whether its received information is actually sent out to other channels after processing. If a channel "gate" closes, its output will always be a constant. In fact, each designed "gate" will have a prior intention to close, unless it has a "strong" duty in sending some of its received information from the input to subsequent layers. We find that implementing this idea in pruning CNNs is unsophisticated, as will be detailed in Sec 4.

Our method neither introduces any extra parameters to the existing CNN, nor changes its computation graph. In fact, it only introduces marginal overheads to existing gradient training of CNNs. It also possess an attractive feature that one can successively build multiple compact models with different inference performances in a single round of resource-intensive training (as in our experiments). This eases the process to choose a balanced model to deploy in production. Probably, the only applicability constraint of our method is that all convolutional layers and fully-connected layer (except the last layer) in the CNN should be batch normalized (Ioffe & Szegedy, 2015). Given batch normalization has becomes a widely adopted ingredient in designing state-of-the-art deep learning models, and many successful CNN models are using it, we believe our approach has a wide scope of potential impacts.[1]

In this paper, we start from rethinking a basic assumption widely explored in existing channel pruning work. We point out several issues and gaps in realizing this assumption successfully. Then, we propose our alternative approach, which works around several numerical difficulties. Finally, we experiment our method across different benchmarks and validate its usefulness and strengths.

## 2 RELATED WORK

Reducing the size of neural network for speeding up its computational performance at inference time has been a long-studied topic in the communities of neural network and deep learning. Pioneer works include Optimal Brain Damage (LeCun et al., 1990) and Optimal Brain Surgeon (Hassibi & Stork, 1993). More recent developments focused on either reducing the structural complexity of a provided network or training a compact or simplified network from scratch. Our work can be categorized into the former, thus the literature review below revolves around reducing the structural complexity.

To reduce the structural complexity of deep learning models, previous work have largely focused on sparsifying the weights of convolutional kernels or the feature maps across multiple layers in a network (Anwar et al., 2017; Han et al., 2015). Some recent efforts proposed to impose structured sparsity on those vector components motivated from the implementation perspective on specialized hardware (Wen et al., 2016; Zhou et al., 2016; Alvarez & Salzmann, 2016; Lebedev & Lempitsky, 2016). Yet as argued by Molchanov et al. (2017), regularization-based pruning techniques require per layer sensitivity analysis which adds extra computations. Their method relies on global rescaling of criteria for all layers and does not require sensitivity estimation, a beneficial feature that our approach also has. To our knowledge, it is also unclear how widely useful those works are in deep learning. In Section 3, we discuss in details the potential issues in regularization-based pruning techniques potentially hurting them being widely applicable, especially for those that regularize high-dimensional tensor parameters or use magnitude-based pruning methods. Our approach works around the mentioned issues by constraining the anticipated pruning operations only to batch-normalized convolutional layers. Instead of posing structured sparsity on kernels or feature maps,

---

[1]For convolution layer which is not originally trained with batch normalization, one can still convert it into a "near equivalent" convolution layer with batch normalization by removing the bias term $b$ and properly setting $\gamma = \sqrt{\sigma + \epsilon}$, $\beta = b + \mu$, where $\sigma$ and $\mu$ are estimated from the outputs of the convolution across all training samples.

we enforce sparsity on the scaling parameter $\gamma$ in batch normalization operator. This blocks the sample-wise information passing through part of the channels in convolution layer, and in effect implies one can safely remove those channels.

A recent work by Huang & Wang (2017) used a similar technique as ours to remove unimportant residual modules in ResNet by introducing extra scaling factors to the original network. However, some optimization subtleties as to be pointed out in our paper were not well explained. Another recent work called *Network-Slimming* (Liu et al., 2017) also aims to sparsify the scaling parameters of batch normalization. But instead of using off-the-shelf gradient learning like theirs, we propose a new algorithmic approach based on ISTA and rescaling trick, improving robustness and speed of the undergoing optimization. In particular, the work of Liu et al. (2017) was able to prune VGG-A model on ImageNet. It is unclear how their work would deal with the $\gamma$-$W$ rescaling effect and whether their approach can be adopted to large pre-trained models, such as ResNets and Inceptions. We experimented with the pre-trained ResNet-101 and compared to most recent work that were shown to work well with large CNNs. We also experimented with an image segmentation model which has an inception-like module (pre-trained on ImageNet) to locate foreground objects.

## 3 RETHINKING THE SMALLER-NORM-LESS-INFORMATIVE ASSUMPTION

In most regularized linear regressions, a large-norm coefficient is often a strong indicator of a highly informative feature. This has been widely perceived in statistics and machine learning communities. Removing features which have a small coefficient does not substantially affect the regression errors. Therefore, it has been an established practice to use tractable norm to regularize the parameters in optimizing a model and pick the important ones by comparing their norms after training. However, this assumption is not unconditional. By using Lasso or ridge regression to select important predictors in linear models, one always has to first normalize each predictor variable. Otherwise, the result might not be explanatory. For example, ridge regression penalizes more the predictors which has low variance, and Lasso regression enforces sparsity of coefficients which are already small in OLS. Such normalization condition for the right use of regularization is often unsatisfied for nonconvex learning. For example, one has to carefully consider two issues outlined below. We provides these two cases to exemplify how regularization could fail or be of limited usage. There definitely exist ways to avoid the described failures.

**Model Reparameterization**. In the first case, we show that it is not easy to have fine-grained control of the weights' norms across different layers. One has to either choose a uniform penalty in all layers or struggle with the reparameterization patterns. Consider to find a deep linear (convolutional) network subject to a least square with Lasso: for $\lambda > 0$,

$$\min_{\{W_i\}_{i=1}^{2n}} \mathbb{E}_{(x,y)\sim\mathcal{D}} \|W_{2n} * \ldots * W_2 * W_1 * x - y\|^2 + \lambda \sum_{i=1}^{n} \|W_{2i}\|_1 \ .$$

The above formulation is not a well-defined problem because for any parameter set $\{W_i\}_{i=1}^{2n}$, one can always find another parameter set $\{W_i'\}_{i=1}^{2n}$ such that it achieves a smaller total loss while keeping the corresponding $l_0$ norm unchanged by actually setting

$$W_i' = \alpha W_i, i = 1, 3, \ldots, 2n - 1 \text{ and } W_i' = W_i/\alpha, i = 2, 4, \ldots, 2n \ ,$$

where $\alpha > 1$. In another word, for any $\epsilon > 0$, one can always find a parameter set $\{W_i\}_{i=1}^{2n}$ (which is usually non-sparse) that minimizes the first least square loss while having its second Lasso term less than $\epsilon$.

We note that gradient-based learning is highly inefficient in exploring such model reparameterization patterns. In fact, there are some recent discussions around this (Dinh et al., 2017). If one adopts a pre-trained model, and augments its original objective with a new norm-based parameter regularization, the new gradient updates may just increase rapidly or it may take a very long time for the variables traveling along the model's reparameterization trajectory. This highlights a theoretical gap questioning existing sparsity-inducing formulation and actual computational algorithms whether they can achieve widely satisfactory parameter sparsification for deep learning models.

**Transform Invariance**. In the second case, we show that batch normalization is not compatible with weight regularization. The example is penalizing $l_1$- or $l_2$-norms of filters in convolution layer which is then followed by a batch normalization: at the $l$-th layer, we let

$$x^{l+1} = \max\{\gamma \cdot \mathrm{BN}_{\mu,\sigma,\epsilon}(W^l * x^l) + \beta, 0\},$$

where $\gamma$ and $\beta$ are vectors whose length is the number of channels. Likewise, one can clearly see that any uniform scaling of $W^l$ which changes its $l_1$- and $l_2$-norms would have no effects on the output $x^{l+1}$. Alternatively speaking, if one is interested in minimizing the weight norms of multiple layers together, it becomes unclear how to choose proper penalty for each layer. Theoretically, there always exists an optimizer that can change the weight to one with infinitesimal magnitude without hurting any inference performance. As pointed by one of the reviewers, one can tentatively avoid this issue by projecting the weights to the surface of unit ball. Then one has to deal with a non-convex feasible set of parameters, causing extra difficulties in developing optimization for data-dependent pruning methods. It is also worth noting that some existing work used such strategy in a layer-by-layer greedy way (He et al., 2017; Zhang et al., 2016).

Based on this discussion, many existing works which claim to use Lasso, group Lasso (e.g. Wen et al. (2016); Anwar et al. (2017)), or thresholding (e.g. Molchanov et al. (2017)) to enforce parameter sparsity have some theoretical gaps to bridge. In fact, many heuristic algorithms in neural net pruning actually do not naturally generate a sparse parameterized solution. More often, thresholding is used to directly set certain subset of the parameters in the network to zeros, which can be problematic. The reason is in essence around two questions. First, by setting parameters less than a threshold to zeros, will the functionality of neural net be preserved approximately with certain guarantees? If yes, then under what conditions? Second, how should one set those thresholds for weights across different layers? Not every layer contributes equally in a neural net. It is expected that some layers act critically for the performance but only use a small computation and memory budget, while some other layers help marginally for the performance but consume a lot resources. It is naturally more desirable to prune calculations in the latter kind of layers than the former.

In contrast with these existing approaches, we focus on enforcing sparsity of a tiny set of parameters in CNN — scale parameter $\gamma$s in all batch normalization. Not only placing sparse constraints on $\gamma$ is simpler and easier to monitor, but more importantly, we have two strong reasons:

1. Every $\gamma$ always multiplies a normalized random variable, thus the channel importance becomes comparable across different layers by measuring the magnitude values of $\gamma$;
2. The reparameterization effect across different layers is avoided if its subsequent convolution layer is also batch-normalized. In other words, the impacts from the scale changes of $\gamma$ parameter are independent across different layers.

Nevertheless, our current work still falls short of a strong theoretical guarantee. We believe by working with normalized feature inputs and their regularized coefficients together, one is closer to a more robust and meaningful approach. Sparsity is not the goal, but to find less important channels using sparsity inducing formulation is.

## 4 Channel Pruning of Batch-Normalized CNN

We describe the basic principle and algorithm of our channel pruning technique.

### 4.1 Preliminaries

**Pruning constant channels**. Consider convolution with batch normalization:

$$x^{l+1} = \max\left\{\gamma^l \cdot \mathrm{BN}_{\mu^l,\sigma^l,\epsilon^l}(W^l * x^l) + \beta^l, 0\right\}.$$

For the ease of notation, we let $\gamma = \gamma^l$. Note that if some element in $\gamma$ is set to zero, say, $\gamma[k] = 0$, its output image $x^{l+1}_{:,:,:,k}$ becomes a constant $\beta_k$, and a convolution of a constant image channel is almost everywhere constant (except for padding regions, an issue to be discussed later). Therefore, we show those constant image channels can be pruned while the same functionality of network is approximately kept:

- If the subsequent convolution layer does not have batch normalization,

$$x^{l+2} = \max \left\{ W^{l+1} * x^{l+1} + b^{l+1}, 0 \right\} \, ,$$

its values (a.k.a. elements in $\beta$) is absorbed into the bias term by the following equation

$$b_{new}^{l+1} := b^{l+1} + I(\gamma = 0) \cdot \text{ReLU}(\beta)^T \text{sum\_reduced}(W_{:,:,:,.}^{l+1}) \, ,$$

such that

$$x^{l+2} \approx \max \left\{ W^{l+1} *_\gamma x^{l+1} + b_{new}^{l+1}, 0 \right\} \, ,$$

where $*_\gamma$ denotes the convolution operator which is only calculated along channels indexed by non-zeros of $\gamma$. Remark that $W^* = \text{sum\_reduced}(W_{:,:,.,.})$ if $W_{a,b}^* = \sum_{i,j} W_{i,j,a,b}$.

- If the subsequent convolution layer has batch normalization,

$$x^{l+2} = \max \left\{ \gamma^{l+1} \cdot \text{BN}_{\mu^{l+1}, \sigma^{l+1}, \epsilon^{l+1}} \left( W^{l+1} * x^{l+1} \right) + \beta^{l+1}, 0 \right\} \, ,$$

instead its moving average is updated as

$$\mu_{new}^{l+1} := \mu^{l+1} - I(\gamma = 0) \cdot \text{ReLU}(\beta)^T \text{sum\_reduced}(W_{:,:,:,.}^{l+1}) \, ,$$

such that

$$x^{l+2} \approx \max \left\{ \gamma^{l+1} \cdot \text{BN}_{\mu_{new}^{l+1}, \sigma^{l+1}, \epsilon^{l+1}} \left( W^{l+1} *_\gamma x^{l+1} \right) + \beta^{l+1}, 0 \right\} \, .$$

Remark that the approximation ($\approx$) is strictly equivalence ($=$) if no padding is used in the convolution operator $*$, a feature that the parallel work Liu et al. (2017) does not possess. When the original model uses padding in computing convolution layers, the network function is not strictly preserved after pruning. In our practice, we fine-tune the pruned network to fix such performance degradation at last. In short, we formulate the network pruning problem as simple as to set more elements in $\gamma$ to zero. It is also much easier to deploy the pruned model, because no extra parameters or layers are introduced into the original model.

To better understand how it works in an entire CNN, imagine a channel-to-channel computation graph formed by the connections between layers. In this graph, each channel is a node, their inference dependencies are represented by directed edges. The $\gamma$ parameter serves as a "dam" at each node, deciding whether let the received information "flood" through to other nodes following the graph. An end-to-end training of channel pruning is essentially like a flood control system. There suppose to be rich information of the input distribution, and in two ways, much of the original input information is lost along the way of CNN inference, and the useful part — that is supposed to be preserved by the network inference — should be label sensitive. Conventional CNN has one way to reduce information: transforming feature maps (non-invertible) via forward propagation. Our approach introduces the other way: block information at each channel by forcing its output being constant using ISTA.

**ISTA**. Despite the gap between Lasso and sparsity in the non-convex settings, we found that ISTA (Beck & Teboulle, 2009) is still a useful sparse promoting method. But we just need to use it more carefully. Specifically, we adopt ISTA in the updates of $\gamma$s. The basic idea is to project the parameter at every step of gradient descent to a potentially more sparse one subject to a proxy problem: let $l$ denote the training loss of interest, at the $(t+1)$-th step, we set

$$\gamma_{t+1} = \min_\gamma \frac{1}{\mu_t} \|\gamma - \gamma_t + \mu_t \nabla_\gamma l_t\|^2 + \lambda \|\gamma\|_1 \, , \tag{1}$$

where $\nabla_\gamma l_t$ is the derivative with respect to $\gamma$ computed at step $t$, $\mu_t$ is the learning rate, $\lambda$ is the penalty. In the stochastic learning, $\nabla_\gamma l_t$ is estimated from a mini-batch at each step. Eq. (1) has closed form solution as

$$\gamma_{t+1} = \text{prox}_{\mu_t \lambda}(\gamma_t - \mu_t \nabla_\gamma l_t) \, ,$$

where $\text{prox}_\eta(x) = \max\{|x| - \eta, 0\} \cdot \text{sgn}(x)$. The ISTA method essentially serves as a "flood control system" in our end-to-end learning, where the functionality of each $\gamma$ is like that of a dam. When $\gamma$ is zero, the information flood is totally blocked, while $\gamma \neq 0$, the same amount of information is passed through in form of geometric quantities whose magnitudes are proportional to $\gamma$.

**Scaling effect**. One can also see that if $\gamma$ is scaled by $\alpha$ meanwhile $W^{l+1}$ is scaled by $1/\alpha$, that is,

$$\gamma := \alpha\gamma, \qquad W^{l+1} := \frac{1}{\alpha}W^{l+1}$$

the output $x^{l+2}$ is unchanged for the same input $x^l$. Despite not changing the output, scaling of $\gamma$ and $W^{l+1}$ also scales the gradients $\nabla_\gamma l$ and $\nabla_{W^{l+1}} l$ by $1/\alpha$ and $\alpha$, respectively. As we observed, the parameter dynamics of gradient learning with ISTA depends on the scaling factor $\alpha$ if one decides to choose it other than 1.0. Intuitively, if $\alpha$ is large, the optimization of $W^{l+1}$ is progressed much slower than that of $\gamma$.

## 4.2 THE ALGORITHM

We describe our algorithm below. The following method applies to both training from scratch or re-training from a pre-trained model. Given a training loss $l$, a convolutional neural net $\mathcal{N}$, and hyper-parameters $\rho, \alpha, \mu_0$, our method proceeds as follows:

1. **Computation of sparse penalty for each layer.** Compute the memory cost per channel for each layer denoted by $\lambda^l$ and set the ISTA penalty for layer $l$ to $\rho\lambda^l$. Here

$$\lambda^l = \frac{1}{I_w^i \cdot I_h^i}\left[ k_w^l \cdot k_h^l \cdot c^{l-1} + \sum_{l' \in \mathcal{T}(l)} k_w^{l'} \cdot k_h^{l'} \cdot c^{l'} + I_w^l \cdot I_h^l \right], \qquad (2)$$

   where
   - $I_w^i \cdot I_h^i$ is the size of input image of the neural network.
   - $k_w^l \cdot k_h^l$ is the kernel size of the convolution at layer $l$. Likewise, $k_w^{l'} \cdot k_h^{l'}$ is the kernel size of subsequent convolution at layer $l'$.
   - $\mathcal{T}(l)$ represents the set of the subsequent convolutional layers of layer $l$
   - $c^{l-1}$ denotes the channel size of the previous layer, which the $l$-th convolution operates over; and $c^{l'}$ denotes the channel size of one subsequent layer $l'$.
   - $I_w^l \cdot I_h^l$ is the image size of the feature map at layer $l$.

2. **$\gamma$-$W$ rescaling trick.** For layers whose channels are going to get reduced, scale all $\gamma^l$s in batch normalizations by $\alpha$ meanwhile scale weights in their subsequent convolutions by $1/\alpha$.

3. **End-to-End training with ISTA on $\gamma$.** Train $\mathcal{N}$ by the regular SGD, with the exception that $\gamma^l$s are updated by ISTA, where the initial learning rate is $\mu_0$. Train $\mathcal{N}$ until the loss $l$ plateaus, the total sparsity of $\gamma^l$s converges, and Lasso $\rho \sum_l \lambda^l \|\gamma^l\|_1$ converges.

4. **Post-process to remove constant channels.** Prune channels in layer $l$ whose elements in $\gamma^l$ are zero and output the pruned model $\widetilde{\mathcal{N}}$ by absorbing all constant channels into subsequent layers (as described in the earlier section.).

5. **$\gamma$-$W$ rescaling trick.** For $\gamma^l$s and weights in $\widetilde{\mathcal{N}}$ which were scaled in Step 2 before training, scale them by $1/\alpha$ and $\alpha$ respectively (scaling back).

6. Fine-tune $\widetilde{\mathcal{N}}$ using regular stochastic gradient learning.

Remark that choosing a proper $\alpha$ as used in Steps 2 and 5 is necessary for using a large $\mu_t \cdot \rho$ in ISTA, which makes the sparsification progress of $\gamma^l$s faster.

## 4.3 GUIDELINES FOR TUNING HYPER-PARAMETERS

We summarize the sensitivity of hyper-parameters and their impacts for optimization below:

- $\mu$ (learning rate): larger $\mu$ leads to fewer iterations for convergence and faster progress of sparsity. But if if $\mu$ too large, the SGD approach wouldn't converge.
- $\rho$ (sparse penalty): larger $\rho$ leads to more sparse model at convergence. If trained with a very large $\rho$, all channels will be eventually pruned.

- $\alpha$ (rescaling): we use $\alpha$ other than 1. only for pretrained models, we typically choose $\alpha$ from $\{0.001, 0.01, 0.1, 1\}$ and smaller $\alpha$ warms up the progress of sparsity.

We recommend the following parameter tuning strategy. First, check the cross-entropy loss and the regularization loss, select $\rho$ such that these two quantities are comparable at the beginning. Second, choose a reasonable learning rate. Third, if the model is pretrained, check the average magnitude of $\gamma$s in the network, choose $\alpha$ such that the magnitude of rescaled $\gamma^l$ is around $100\mu\lambda^l\rho$. We found as long as one choose those parameters in the right range of magnitudes, the optimization progress is enough robust. Again one can monitor the mentioned three quantities during the training and terminate the iterations when all three quantities plateaus.

There are several patterns we found during experiments that may suggest the parameter tuning has not been successful. If during the first few epochs the Lasso-based regularization loss keeps decreasing linearly while the sparsity of $\gamma$s stays near zero, one may decrease $\alpha$ and restart. If during the first few epochs the sparsity of $\gamma$s quickly raise up to 100%, one may decrease $\rho$ and restart. If during the first few epochs the cross-entropy loss keeps at or increases dramatically to a non-informative level, one may decrease $\mu$ or $\rho$ and restart.

## 5 EXPERIMENTS

### 5.1 CIFAR-10 EXPERIMENT

We experiment with the standard image classification benchmark CIFAR-10 with two different network architectures: ConvNet and ResNet-20 (He et al., 2016). We resize images to $32 \times 32$ and zero-pad them to $40 \times 40$. We pre-process the padded images by randomly cropping with size $32 \times 32$, randomly flipping, randomly adjusting brightness and contrast, and standardizing them such that their pixel values have zero mean and one variance.

**ConvNet** For reducing the channels in ConvNet, we are interested in studying whether one can easily convert a over-parameterized network into a compact one. We start with a standard 4-layer convolutional neural network whose network attributes are specified in Table 1. We use a fixed learning rate $\mu_t = 0.01$, scaling parameter $\alpha = 1.0$, and set batch size to 125.

Model A is trained from scratch using the base model with an initial warm-up $\rho = 0.0002$ for 30k steps, and then is trained by raising up $\rho$ to 0.001. After the termination criterion are met, we prune the channels of the base model to generate a smaller network called model A. We evaluate the classification performance of model A with the running exponential average of its parameters. It is found that the test accuracy of model A is even better than the base model. Next, we start from the pre-trained model A to create model B by raising $\rho$ up to 0.002. We end up with a smaller network called model B, which is about 1% worse than model A, but saves about one third parameters. Likewise, we start from the pre-trained model B to create model C. The detailed statistics and its pruned channel size are reported in Table 1. We also train a reference ConvNet from scratch whose channel sizes are 32-64-64-128 with totally 224,008 parameters and test accuracy being 86.3%. The referenced model is not as good as Model B, which has smaller number of parameters and higher accuracy.

We have two major observations from the experiment: (1) When the base network is over-parameterized, our approach not only significantly reduces the number of channels of the base model but also improves its generalization performance on the test set. (2) Performance degradation seems unavoidable when the channels in a network are saturated, and our approach gives satisfactory trade-off between test accuracy and model efficiency.

**ResNet-20** We also want to verify our second observation with the state-of-art models. We choose the popular ResNet-20 as our base model for the CIFAR-10 benchmark, whose test accuracy is 92%. We focus on pruning the channels in the residual modules in ResNet-20, which has 9 convolutions in total. As detailed in Table 2, model A is trained from scratch using ResNet-20's network structure as its base model. We use a warm-up $\rho = 0.001$ for 30k steps and then train with $\rho = 0.005$. We are able to remove 37% parameters from ResNet-20 with only about 1 percent accuracy loss. Likewise, Model B is created from model A with a higher penalty $\rho = 0.01$.

| | | | base | model A | model B | model C |
|---|---|---|---|---|---|---|
| layer | output | kernel | channel | channel | channel | channel |
| conv1 | $32 \times 32$ | $5 \times 5$ | 96 | 53 | 41 | 31 |
| pool1 | $16 \times 16$ | $3 \times 3$ | | | | |
| conv2 | $16 \times 16$ | $5 \times 5$ | 192 | 86 | 64 | 52 |
| pool2 | $8 \times 8$ | $3 \times 3$ | | | | |
| conv3 | $8 \times 8$ | $3 \times 3$ | 192 | 67 | 52 | 40 |
| pool4 | $4 \times 4$ | $3 \times 3$ | | | | |
| fc | $1 \times 1$ | $4 \times 4$ | 384 | 128 | 128 | 127 |
| $\rho$ | | | | 0.001 | 0.002 | 0.008 |
| param. size | | | 1,986,760 | 309,655 | 207,583 | 144,935 |
| test accuracy (%) | | | 89.0 | 89.5 | 87.6 | 86.0 |

Table 1: Comparisons between different pruned networks and the base network.

| | group - block | 1-1 | 1-2 | 1-3 | 2-1 | 2-2 | 2-3 | 3-1 | 3-2 | 3-3 |
|---|---|---|---|---|---|---|---|---|---|---|
| ResNet-20 | channels param size. = 281,304 test accuracy (%) = 92.0 | 16 | 16 | 16 | 32 | 32 | 32 | 64 | 64 | 64 |
| model A | channels param size. = 176,596 test accuracy (%) = 90.9 | 12 | 6 | 11 | 32 | 28 | 28 | 47 | 34 | 25 |
| model B | channels param size. = 90,504 test accuracy (%) = 88.8 | 8 | 2 | 7 | 27 | 18 | 16 | 25 | 9 | 8 |

Table 2: Comparisons between ResNet-20 and its two pruned versions. The last columns are the number of channels of each residual modules after pruning.

## 5.2 ILSVRC2012 EXPERIMENT

We experiment our approach with the pre-trained ResNet-101 on ILSVRC2012 image classification dataset (He et al., 2016). ResNet-101 is one of the state-of-the-art network architecture in ImageNet Challenge. We follow the standard pipeline to pre-process images to $224 \times 224$ for training ResNets. We adopt the pre-trained TensorFlow ResNet-101 model whose single crop error rate is 23.6% with about $4.47 \times 10^7$ parameters.[2] We set the scaling parameter $\alpha = 0.01$, the initial learning rate $\mu_t = 0.001$, the sparsity penalty $\rho = 0.1$, and the batch size $= 128$ (across 4 GPUs). The learning rate is decayed every four epochs with rate $0.86$. We create two pruned models from the different iterations of training ResNet-101: one has $2.36 \times 10^7$ parameters and the other has $1.73 \times 10^7$ parameters. We then fine-tune these two models using the standard way for training ResNet-101, and report their error rates. The Top-5 error rate increases of both models are less than $0.5\%$. The Top-1 error rates are summarized in Table 3. To our knowledge, only a few works have reported their performance on this very large-scale benchmark w.r.t. the Top-1 errors. We compare our approach with some recent works in terms of model parameter size, flops, and error rates. As shown in Table 3, our model v2 has achieved a compression ratio more than 2.5 while maintaining more than 1% lower error rates than that of other state-of-the-art models at comparable size of parameters.

In the first experiment (CIFAR-10), we train the network from scratch and allocate enough steps for both $\gamma$ and $W$ adjusting their own scales. Thus, initialization of an improper scale of $\gamma$-$W$ is not really an issue given we optimize with enough steps. But for the pre-trained models which were originally optimized without any constraints of $\gamma$, the $\gamma$s scales are often unanticipated. It actually takes as many steps as that of training from scratch for $\gamma$ to warm up. By adopting the rescaling trick setting $\alpha$ to a smaller value, we are able to skip the warm-up stage and quick start to sparsify $\gamma$s. For example, it might take more than a hundred epoch to train ResNet-101, but it only takes about 5-10 epochs to complete the pruning and a few more epochs to fine-tune.

---

[2]https://github.com/tensorflow/models/tree/master/slim

| network | param size. | flops | error (%) | ratio |
|---|---|---|---|---|
| resnet-50 pruned (Huang & Wang, 2017) | $\sim 1.65 \times 10^7$ | $3.03 \times 10^9$ | $\sim 26.8$ | 66% |
| resnet-101 pruned (v2, ours) | $1.73 \times 10^7$ | $3.69 \times 10^9$ | **25.44** | 39% |
| resnet-34 pruned (Li et al., 2017) | $1.93 \times 10^7$ | $2.76 \times 10^9$ | 27.8 | 89% |
| resnet-34 | $2.16 \times 10^7$ | $3.64 \times 10^9$ | 26.8 | - |
| resnet-101 pruned (v1, ours) | $2.36 \times 10^7$ | $4.47 \times 10^9$ | **24.73** | 53% |
| resnet-50 | $2.5 \times 10^7$ | $4.08 \times 10^9$ | 24.8 | - |
| resnet-101 | $4.47 \times 10^7$ | $7.8 \times 10^9$ | 23.6 | - |

Table 3: Attributes of different versions of ResNet and their single crop errors on ILSVRC2012 benchmark. The last column means the parameter size of pruned model vs. the base model.

## 5.3 IMAGE FOREGROUND-BACKGROUND SEGMENTATION EXPERIMENT

As we have discussed about the two major observations in Section 5.1, a more appealing scenario is to apply our approach in pruning channels of over-parameterized model. It often happens when one adopts a pre-trained network on a large task (such as ImageNet classification) and fine-tunes the model to a different and smaller task (Molchanov et al., 2017). In this case, one might expect that some channels that have been useful in the first pre-training task are not quite contributing to the outputs of the second task.

We describe an image segmentation experiment whose neural network model is composed from an inception-like network branch and a densenet network branch. The entire network takes a $224 \times 224$ image and outputs binary mask at the same size. The inception branch is mainly used for locating the foreground objects while the densenet network branch is used to refine the boundaries around the segmented objects. This model was originally trained on multiple datasets.

In our experiment, we attempt to prune channels in both the inception branch and densenet branch. We set $\alpha = 0.01$, $\rho = 0.5$, $\mu_t = 2 \times 10^{-5}$, and batch size = 24. We train the pre-trained base model until all termination criterion are met, and build the pruned model for fine-tuning. The pruned model saves 86% parameters and 81% flops of the base model. We also compare the fine-tuned pruned model with the pre-trained base model across different test benchmark. Mean IOU is used as the evaluation metric.[3] It shows that pruned model actually improves over the base model on four of the five test datasets with about $2\% \sim 5\%$, while it performs worse than the base model on the most challenged dataset DUT-Omron, whose foregrounds might contain multiple objects.

| | base model | pruned model |
|---|---|---|
| test dataset (#images) | mIOU | mIOU |
| MSRA10K (Liu et al., 2011) (2,500) | 83.4% | 85.5% |
| DUT-Omron (Yang et al., 2013) (1,292) | 83.2% | 79.1% |
| Adobe Flickr-portrait (Shen et al., 2016) (150) | 88.6% | 93.3% |
| Adobe Flickr-hp (Shen et al., 2016) (300) | 84.5% | 89.5% |
| COCO-person (Lin et al., 2014) (50) | 84.1% | 87.5% |
| param. size | $1.02 \times 10^7$ | $1.41 \times 10^6$ |
| flops | $5.68 \times 10^9$ | $1.08 \times 10^9$ |

Table 4: mIOU reported on different test datasets for the base model and the pruned model.

## 6 CONCLUSIONS

We proposed a model pruning technique that focuses on simplifying the computation graph of a deep convolutional neural network. Our approach adopts ISTA to update the $\gamma$ parameter in batch normalization operator embedded in each convolution. To accelerate the progress of model pruning, we use a $\gamma$-$W$ rescaling trick before and after stochastic training. Our method cleverly avoids some possible numerical difficulties such as mentioned in other regularization-based related work, hence

---

[3] https://www.tensorflow.org/api_docs/tf/metrics/mean_iou

is easier to apply for practitioners. We empirically validated our method through several benchmarks and showed its usefulness and competitiveness in building compact CNN models.

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

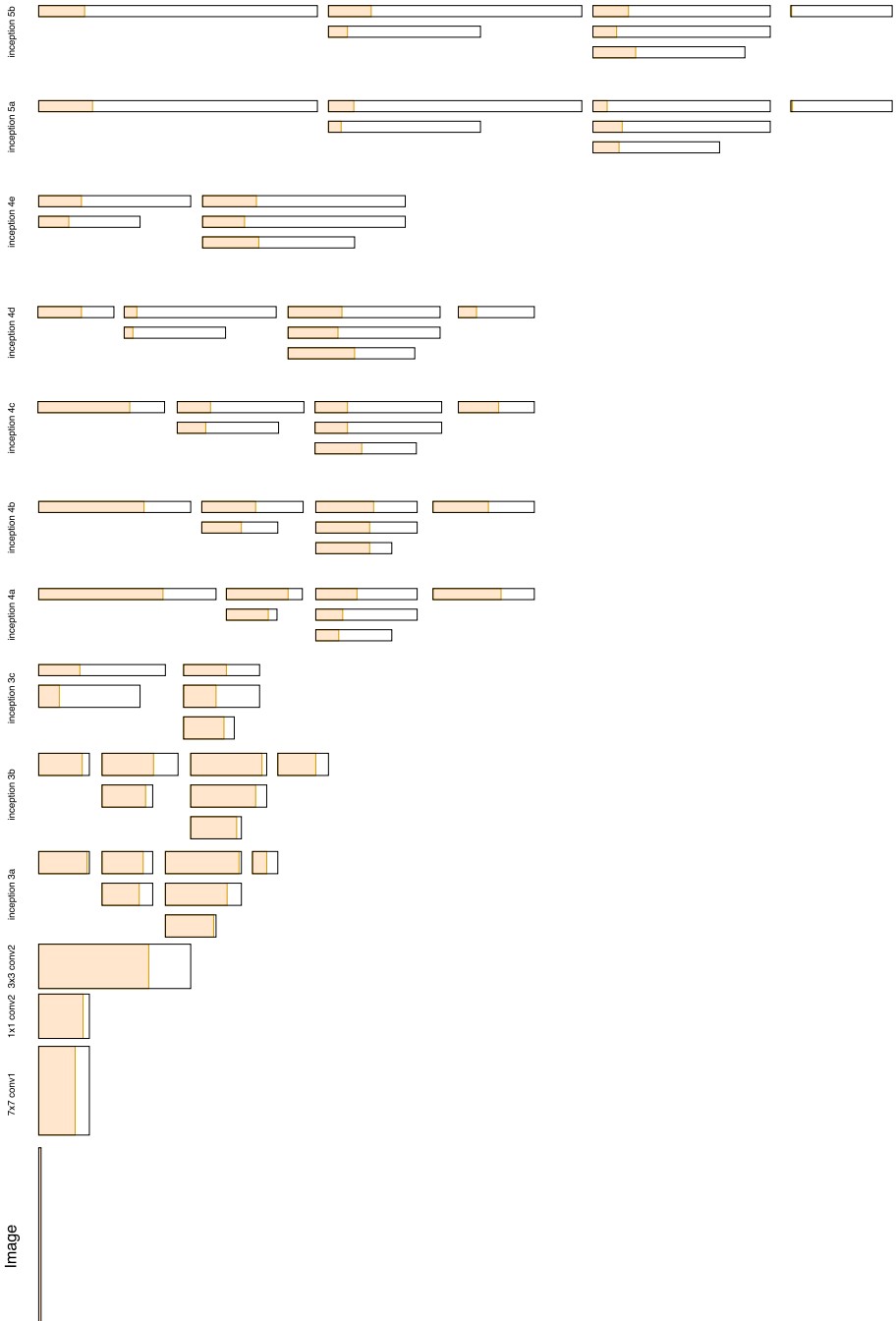

Figure 1: Visualization of the number of pruned channels at each convolution in the inception branch. Colored regions represents the number of channels kept. The height of each bar represents the size of feature map, and the width of each bar represents the size of channels. It is observed that most of channels in the bottom layers are kept while most of channels in the top layers are pruned.

