# OpenReview forum: "Rethinking the Smaller-Norm-Less-Informative Assumption in Channel Pruning of Convolution Layers"
_ICLR.cc/2018/Conference — Accept (Poster)_

### Official Review · AnonReviewer2 · 2017-11-26
**Data-dependent channel pruning approach to simplify CNNs. Right questions but missing answers.**

**Rating:** 5
**Confidence:** 5

**Review:**

In this paper, the authors propose a data-dependent channel pruning approach to simplify CNNs with batch-normalizations. The authors view CNNs as a network flow of information and applies sparsity regularization on the batch-normalization scaling parameter \gamma which is seen as a “gate” to the information flow. Specifically, the approach uses iterative soft-thresholding algorithm step to induce sparsity in \gamma during the overall training phase of the CNN (with additional rescaling to improve efficiency. In the experiments section, the authors apply their pruning approach on a few representative problems and networks.

The concept of applying sparsity on \gamma to prune channels is an interesting one, compared to the usual approaches of sparsity on weights. However, the ISTA, which is equivalent to L1 penalty on \gamma is in spirit same as “smaller-norm-less-informative” assumption. Hence, the title seems a bit misleading.

The quality and clarity of the paper can be improved in some sections. Some specific comments by section:

3. Rethinking Assumptions:
-	While both issues outlined here are true in general, the specific examples are either artificial or can be resolved fairly easily. For example: L-1 norm penalties only applied on alternate layers is artificial and applying the penalties on all Ws would fix the issue in this case. Also, the scaling issue of W can be resolved by setting the norm of W to 1, as shown in He et. al., 2017. Can the authors provide better examples here?
-	Can the authors add specific citations of the existing works which claim to use Lasso, group Lasso, thresholding to enforce parameter sparsity?

4. Channel Pruning
-	The notation can be improved by defining or replacing “sum_reduced”
-	ISTA – is only an algorithm, the basic assumption is still L1 -> sparsity or smaller-norm-less-informative. Can the authors address the earlier comment about “a theoretical gap questioning existing sparsity inducing formulation and actual computational algorithms”?
-	Can the authors address the earlier comment on “how to set thresholds for weights across different layers”, by providing motivation for choice of penalty for each layer?
-	Can the authors address the earlier comment on how their approach provides “guarantees for preserving neural net functionality approximately”?

5. Experiments
-	CIFAR-10: Since there is loss of accuracy with channel pruning, it would be useful to compare accuracy of a pruned model with other simpler models with similar param.size? (like pruned-resnet-101 vs. resnet-50 in ISLVRC subsection)
-	ISLVRC: The comparisons between similar param-size models is exteremely useful in highlighting the contribution of this. However, resnet-34/50/101 top-1 error rates from Table 3/4 in (He et.al. 2016) seem to be lower than reported in table 3 here. Can the authors clarify?
-	Fore/Background: Can the authors add citations for datasets, metrics for this problem?


Overall, the channel pruning with sparse \gammas is an interesting concept and the numerical results seem promising. The authors have started with right motivation and the initial section asks the right questions, however, some of those questions are left unanswered in the subsequent work as detailed above.

---

> ### Author Response · Authors · 2017-12-11
> **Responses to Review3**
>
> Thanks for your thoughtful review. We have given serious considerations of your concerns and revise our manuscript to accommodate your suggestions. Please see the details below.
>
> 1. "However, the ISTA, which is equivalent to L1 penalty on \gamma is in spirit same as “smaller-norm-less-informative” assumption. Hence, the title seems a bit misleading. "
>
> ISTA solving L1 regularization problem has been theoretically justified only for strongly convex problems. It is however a heuristic when applying to neural networks. We generally treat it as a sparsity promoting method, rather than a solid optimization tool.
>
> Our paper is not to challenge this assumption, but wants to emphasize when this assumption could be valid. Even for linear models, this assumption is questionable if predictor variables are not normalized ahead of time. We believe this is also the case for other regularized channel selection problems.
>
> We have made our point clearer in the revision. (see Sec 3., the first paragraph)
>
>
> 2. Q3.1
> "… the specific examples are either artificial or can be resolved fairly easily."
>
> We provides these two cases to exemplify how regularization could fail or be of limited usage. There definitely exist ways to avoid the specific failures. In the first case, we show that it is not easy to have fine-grained control of the Ws’ norms across different layers. One has to either choose a uniform penalty for Ws in all layers or struggle with the reparameterization patterns. In the second case, we show that batch normalization is not compatible with W regularization. By constraining W to norm 1.,  one has to deal with a non-convex feasible set of parameters, causing extra difficulties in developing optimization for data-dependent pruning method. (He et. al., 2017) is a data independent approach.
>
> 3. Q3.2
> "Can the authors add specific citations of the existing works which claim to use Lasso, group Lasso, thresholding to enforce parameter sparsity?"
>
> We have added related citations in the revision.
>
>
>
> 4. Q4.1
> "The notation can be improved by defining or replacing “sum_reduced”"
>
> We added the definition of relevant notation.
>
>
>
> 5. Q4.2
> "Can the authors address the earlier comment about “a theoretical gap questioning existing sparsity inducing formulation and actual computational algorithms”?"
>
> Our current work still falls short of a strong theoretical guarantee. But we believe by working with normalized feature inputs and their regularized coefficients together, one is closer to a more robust and meaningful approach. Sparsity is not the goal. The goal is to find less important channels using sparsity inducing formulation.
>
>
> 6. Q4.3
> "Can the authors address the earlier comment on “how to set thresholds for weights across different layers”, by providing motivation for choice of penalty for each layer?"
>
> In our method, we set the penalty for each layer to be proportional to the memory needed per channel. Thanks to the benefits of batch normalization, we could have this fine-grained control of penalties. (see Algorithm Step 1.)
>
>
> 7. Q4.4
> "Can the authors address the earlier comment on how their approach provides “guarantees for preserving neural net functionality approximately”?"
>
> As we mentioned ISTA is used as a heuristic in training the network and selecting channels to be pruned. Once this step is done, we do have guarantee that after pruning all channels whose gammas are zero, the functionality of CNN is identically kept if no padding is used.
>
>
>
> 8. Q5.1
> "CIFAR-10: Since there is loss of accuracy with channel pruning, it would be useful to compare accuracy of a pruned model with other simpler models with similar param.size?"
>
> We do have this comparison done, which were not included in the initial submission. We have included it in the revision. “We also train a reference ConvNet from scratch whose channel sizes are 32-64-64-128 with totally 224,008 parameters and test accuracy being 86.3\%. The referenced model is not as good as Model B, which has smaller number of parameters and higher accuracy. ”
>
>
> 9. Q5.2
> "ISLVRC: … However, resnet-34/50/101 top-1 error rates from Table 3/4 in (He et.al. 2016) seem to be lower than reported in table 3 here. Can the authors clarify?"
>
> In (He et al. 2016), the error is evaluated based on ensembling 10 random crops of a test image. Our number reported is based on a single crop, which has been used as a practice in other related work.
>
>
>
> 10. Q5.3
> "Fore/Background: Can the authors add citations for datasets, metrics for this problem?"
>
> We added in the revision. Note some labels are originally created/collected by researchers of the company, which have not yet been released. We use the mean IOU ( https://www.tensorflow.org/api_docs/python/tf/metrics/mean_iou ) to evaluate the model.

---

### Official Review · AnonReviewer3 · 2017-11-27
**Review for Rethinking the Smaller-Norm-Less-Informative Assumption in Channel Pruning of Convolution Layers**

**Rating:** 7
**Confidence:** 3

**Review:**

This paper is well written and it was easy to follow. The authors propose prunning model technique by enforcing sparsity on the scaling parameter of batch normalization layers. This is achieved by forcing the output of some channels being constant during training. This is achieved an adaptation of ISTA algorithm to update the batch-norm parameter.

The authors evaluate the performance of the proposed approach on different classification and segmentation tasks. The method seems to be relatively straightforward to train and achieve good performance (in terms of performance/parameter reduction) compared to other methods on Imagenet.

Some of the hyperparameters used (alpha and specially rho) seem to be used very ad-hoc. Could the authors explain their choices? How sensible is the algorithm to these hyperparameters?
It would be nice to see empirically how much of computation the proposed approach takes during training. How much longer does it takes to train the model with the ISTA based constraint?

Overall this is a good paper and I believe it should be accepted, given the authors are more clear on the details pointed above.

---

> ### Author Response · Authors · 2017-12-11
> **Responses to Review2**
>
> Thanks for your review. Please see below how we address your concerns in the revision.
>
> 1. "Some of the hyperparameters used (alpha and specially rho) seem to be used very ad-hoc. Could the authors explain their choices? How sensible is the algorithm to these hyperparameters?"
>
> There are several hyperparameters one has to carefully choose, and they have a mixed effect on the number of iterations needed and model performance. In the revision, we include a section (Sec 4.3) guiding one to properly tune those parameters.
>
> Here is what we have summarized:
>
> \mu (learning rate): larger \mu leads to fewer iterations for convergence and faster progress of sparsity, but if \mu is too large, the SGD approach wouldn’t converge.
>
> \rho (sparse penalty): larger \rho leads to more sparse model at convergence. If trained with a very large \rho, all channels will be eventually pruned.
>
> \alpha (rescaling): we use \alpha other than one only for pretrained models, we usually choose \alpha from {0.001, 0.01, 0.1, 1} and larger \alpha slows down the progress of sparsity.
>
>
>
> 2. "It would be nice to see empirically how much of computation the proposed approach takes during training. How much longer does it takes to train the model with the ISTA based constraint?"
>
> Per-iteration computation is almost the same as regular SGD. For train-from-scratch setting, the number of iterations is roughly the same as the number one needs for SGD. For pre-trained ImageNet model (which is typically trained with a hundred epochs), our method takes about 5~10 epochs for ISTA-based training, and takes another few epochs for regular fine-tuning. We have included this in the revision.

---

### Official Review · AnonReviewer1 · 2017-11-28
**A pruning approach based on the batch normalization layer. The algorithm is easy to reproduce and seems to obtain interesting results. Sensibility analysis would be nice**

**Rating:** 6
**Confidence:** 5

**Review:**

This paper proposes an interesting  approach to prune a deep model from a computational point of view. The idea is quite simple as pruning using the connection in the batch norm layer. It is interesting to add the memory cost per channel into the optimization process.

The paper suggests normal pruning does not necessarily preserve the network function. I wonder if this is also applicable to the proposed method and how can this be evidenced.

As strong points, the paper is easy to follow and does a good review of existing methods. Then, the proposal is simple and easy to reproduce and leads to interesting results. It is clearly written (there are some typos / grammar errors).

As weak points:
1) The paper claims the selection of \alpha is critical but then, this is fixed empirically without proper sensitivity analysis. I would like to see proper discussion here. Why is \alpha set to 1.0 in the first experiment while set to a different number elsewhere.

2) how is the pruning (as post processing) performed for the base model (the so called model A).

In section 4, in the algorithmic steps. How does the 4th step compare to the statement in the initial part of the related work suggesting zeroed-out parameters can affect the functionality of the network?

3) Results for CIFAR are nice although not really impressive as the main benefit comes from the fully connected layer as expected.

---

> ### Author Response · Authors · 2017-12-11
> **Responses to Review1**
>
> Thanks for your review. It helps us to prepare the revision. We want to address all your concerns in the revision as below.
>
> 1. "The paper suggests normal pruning does not necessarily preserve the network function. I wonder if this is also applicable to the proposed method and how can this be evidenced. "
>
> Our approach enforces sparsity of scaling parameters in BN, sharing a similar spirit of regularized linear models. By using Lasso/ridge regression to select important predictors in linear models, one always has to first normalize each predictor variable. Otherwise, the result might not be explanatory anyway. For example, Ridge penalizes more the predictors which have low variance, and Lasso likely enforces sparsity of coefficients which are already small in OLS. [1]
>
> Those issues are widely considered by statisticians, and we believe it also happens in  the neural networks. We were not trying to prove our method should be superior, but rather want to remind the community that small magnitude is not necessarily a good reason to prune. Combining with some additional assumptions (e.g. normalized predictors, or in our case, using BN) might be a plausible excuse. We are interested to see if there is any supporting theoretical guarantee in the future.
>
> 2. "The paper claims the selection of \alpha is critical but then, this is fixed empirically without proper sensitivity analysis. I would like to see proper discussion here. Why is \alpha set to 1.0 in the first experiment while set to a different number elsewhere. "
>
> In the first experiment (CIFAR-10), we train the network from scratch and allocate enough steps for both $\gamma$ and $W$ to adjust  their own scales. Thus, initialization of an improper scale of $\gamma$-$W$ is not really an issue given we optimize with enough steps.
>
> But for the pre-trained models which were originally optimized without any constraint of $\gamma$, the $\gamma$’s scales are often unanticipated (compared to learning rate and penalty). It actually takes as many steps as that of training from scratch for $\gamma$ to warm up. By adopting the rescaling trick setting $\alpha$ to a smaller value, we are able to skip the warm-up stage and quick start to sparsify $\gamma$s. For example, it might take more than a hundred epochs to train ResNet-101, but it only takes about 5-10 epochs to complete the pruning and a few more epochs to fine-tune.
> We have added those details in the revision.
>
>
> 3. "How is the pruning (as post processing) performed for the base model (the so called model A). In section 4, in the algorithmic steps. How does the 4th step compare to the statement in the initial part of the related work suggesting zeroed-out parameters can affect the functionality of the network?"
>
> By the end of stochastic training, some gamma will stay at (exact) zero by the nature of ISTA. We then work on removing the channels with those zero-gamma from the base model (so-called post-processing) using the method described in Sec 4.1. Previous work adopting regularization method still needs to select a threshold for each layer’s parameters in order to be zeroed out. Zero-out small magnitude parameters changes the functionality of CNN, but our approach keeps the identical functionality of CNN after post-processing, if there is no padding in the base model.
>
>
> 4. "Results for CIFAR are nice although not really impressive as the main benefit comes from the fully connected layer as expected."
>
> Our experimented two base models (CNN and ResNet-20) for CIFAR-10 are convolutional networks with all layers being convolutions (except the last one). We have reported the number of channels in convolution layer that have been pruned in the paper (See Table 1 & 2). The result suggests our method is able to significantly reduce the channel size for each convolution layer without sacrificing a lot performance. For example, comparing Model A with Model B/C, the pruning mainly happened at convolution layers. For ResNet-20 experiment, we only attempt to prune convolutions in the residual modules, the last layer is an average pooling (with no parameters).
>
>
>
> [1] Hastie, T et al. The Elements of Statistical Learning, Springer.

---

### Public Comment · ~Zhuang_Liu1 · 2017-12-02
**Clarification and Comparison with [1]**

Hi authors, nice paper, well done! In particular, I think the rationale for using Batch Normalization's \gamma scaling factors as the pruning criterion is explained in a very clear fashion.

However, given this "sparsify BN's \gamma and prune channels accordingly" technique is already proposed by the ICCV paper [1], I think it would be great to include an experimental comparison. In my understanding, the essential difference is that you use ISTA to sparsify \gamma instead of L1 regularization as in [1].

Moreover, I think the "parallel work" argument might not be fully valid, as [1] was published and presented before ICLR submission.

Nevertheless, I believe the paper is still a good contribution to ICLR if [1] is cited as prior work (rather than parallel work), and a comparison is conducted.

[1] Liu et al. ICCV 2017. Learning Efficient Convolutional Networks through Network Slimming.  https://arxiv.org/abs/1708.06519

---

> ### Author Response · Authors · 2017-12-11
> **Response and Clarifications**
>
> Hi Zhuang and your collaborators,
>
> Thanks for your comment!
>
> In the revision, we have acknowledged that [1] is a prior work, but it has been available on arXiv after we fully developed our method in the summer. We noticed [1] during its presentation at ICCV 2017 (occurred in late October). We felt obliged to cite the work in our submission, as we all agreed it is a very relevant paper. We were sincerely not aware of the arXiv version posted on late August.
>
> Although the basic idea “enforcing sparsity on BN’s gamma” was firstly discovered in the work of [1], there are several differences in techniques and experiments that we want to highlight here in order to justify our work makes a significant contribution to this new concept.
>
> 1. As mentioned by your comments, the major difference is we use ISTA rather than regularization to enforce the sparsity of BN’s gamma. The use of ISTA enables to explicitly monitor the sparsification progress of gammas. As we explained in the revised version, this helps one to effectively tune the hyper-parameters.
>
> The ISTA when applied to linear models is a standard technique for handling regularization. But there has been no theoretical guaranteed for its use in non-convex problems. In fact, we use ISTA in our work as a sparsity promoting heuristic.
>
>
> 2. Based on our understanding, the mentioned "sparsify BN's \gamma and prune channels accordingly" technique we use is not exactly the same as one used by authors of [1].
>
> According to [1], their method “... prune channels with near-zero scaling factors, by removing all their incoming and outgoing connections and corresponding weights.”  and “Pruning may temporarily lead to some accuracy loss, when the pruning ratio is high. But this can be largely compensated by the followed fine-tuning process on the pruned network.”
>
> In our method of ISTA, some gammas will stay exactly at zero (rather than near zero) after training. We prune channels with zero scaling factors. In addition to remove connections as they did, we also adjust the bias term in the follow-up layers to compensate the effects of removing BN’s beta. By doing such procedures, the network functionality is exactly preserved if there is no padding in convolution. (See Sec. 4.1)
>
>
> 3. The gamma-W rescaling effects are not explicitly addressed in [1]. We feel it is an important issue to be addressed in order to use this new technique for large pre-trained models.
>
> In our experiments, we discovered that if gamma is initially not properly scaled before training, it takes a lot more steps for them to be sparse. Such cases could happen because the original pre-trained model was not trained with any constraints on gammas. Without explicitly address this issue, It could be impractical for applying the method to many large pre-trained model as we experimented with ImageNet challenge. In the ImageNet benchmark of [1], they experimented their method on VGG-A. It is unclear how much computation [1] could take to prune larger models (e.g. ResNets or Inceptions on ImageNet).
>
>
>
> 4. We essentially hold a quite different set of rationales and motivations for proposing the concept from those were argued in [1]. We feel encouraged that our thoughts have been recognized by you as well as three other reviewers. Besides the techniques and empirical evidences developed in our paper, we believe those discussions will also be good additions to the field.
>
> We have accommodated above discussions in our revision.

---

> > ### Public Comment · ~Zhuang_Liu1 · 2017-12-14
> > **Thanks**
> >
> > Hi! Thanks for your response and the revision.  With your clarifications, now the difference is made more clear, and we highly acknowledge your contributions.

---

### Author Response · Authors · 2018-10-16
**code available**

checkout at

https://github.com/bobye/batchnorm_prune/tree/master/src/python/cifar10

---

### Decision · Program_Chairs · 2018-01-29
**ICLR 2018 Conference Acceptance Decision**

**Decision:**

Accept (Poster)

**Comment:**

The paper received scores either side of the borderline: 6 (R1), 5 (R2), 7 (R3). R1 and R3 felt the idea to be interesting, simple and effective. R2 raised a number of concerns which the rebuttal addressed satisfactorily. Therefore the AC feels the paper can be accepted.